# A System for General In-Hand Object Re-Orientation

**Tao Chen, Jie Xu, Pulkit Agrawal**
Massachusetts Institute of Technology
`{taochen, jiex, pulkitag}@mit.edu`

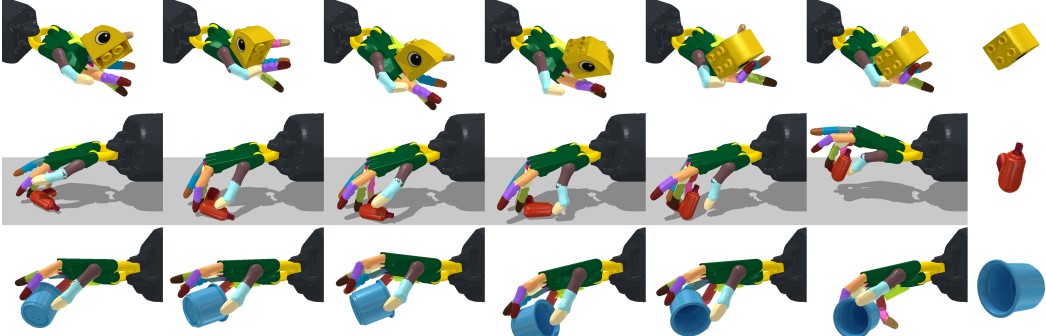

Figure 1: We present a simple framework for learning policies for reorienting a large number of objects in scenarios where the (1) hand faces upward, (2) hand faces downward with a table below the hand and (3) without the support of the table. The object orientation in the rightmost image in each row shows the target orientation.

**Abstract:** In-hand object reorientation has been a challenging problem in robotics due to high dimensional actuation space and the frequent change in contact state between the fingers and the objects. We present a simple model-free framework that can learn to reorient objects with both the hand facing upwards and downwards. We demonstrate the capability of reorienting over 2000 geometrically different objects in both cases. The learned policies show strong zero-shot transfer performance on new objects. We provide evidence that these policies are amenable to real-world operation by distilling them to use observations easily available in the real world. The videos of the learned policies are available at: https://taochenshh.github.io/projects/in-hand-reorientation.

**Keywords:** Dexterous manipulation, in-hand manipulation, object reorientation

## 1 Introduction

A common maneuver in many tasks of daily living is to pick an object, reorient it in hand and then either place it or use it as a tool. Consider three simplified variants of this maneuver shown in Figure 1. The task in the top row requires an upward-facing multi-finger hand to manipulate an *arbitrary* object in a random orientation to a goal configuration shown in the rightmost column. The next two rows show tasks where the hand is facing downward and is required to reorient the object either using the table as a support or without the aid of any support surface respectively. The last task is the hardest because the object is in an intrinsically unstable configuration owing to the downward gravitational force and lack of support from the palm. Additional challenges in performing such manipulation with a multi-finger robotic hand stem from the control space being high-dimensional and reasoning about the multiple transitions in the contact state between the finger and the object. Due to its practical utility and several unsolved issues, in-hand object reorientation remains an active area of research.

Past work has tackled the in-hand reorientation problem via several approaches: (i) The use of analytical models with powerful trajectory optimization methods [1, 2, 3]. While these methods demonstrated remarkable performance, the results were largely in simulation with simple object geometries and required detailed knowledge of the object model and physical parameters. As such, it remains unclear how to scale these methods to real-world and generalize to new objects. Another line of work has employed (ii) model-based reinforcement learning [4, 5]; or (iii) model-free reinforcement learning with [6, 7, 8, 9] and without expert demonstrations [10, 11, 12, 13]. While some of these

5th Conference on Robot Learning (CoRL 2021), London, UK.

works demonstrated learned skills on real robots, it required use of additional sensory apparatus not readily available in the real-world (e.g., motion capture system) to infer the object state, and the learned policies did not generalize to diverse objects. Furthermore, most prior methods operate in the simplified setting of the hand facing upwards. The only exception is pick-and-place, but it does not involve any in-hand re-orientation. A detailed discussion of prior research is provided in Section 5.

In this paper, our goal is to study the object reorientation problem with a multi-finger hand in its general form. We desire (a) manipulation with hand facing upward or downward; (b) the ability of using external surfaces to aid manipulation; (c) the ability to reorient objects of novel shapes to arbitrary orientations; (d) operation from sensory data that can be easily obtained in the real world such as RGBD images and joint positions of the hand. While some of these aspects have been individually demonstrated in prior work, we are unaware of any published method that realizes all four. Our main contribution is building a system that achieves the desiderata. The core of our framework is a model-free reinforcement learning with three key components: teacher-student learning, gravity curriculum, and stable initialization of objects. Our system requires no knowledge of object or manipulator models, contact dynamics or any special pre-processing of sensory observations. We experimentally test our framework using a simulated Shadow hand. Due to the scope of the problem and the ongoing pandemic, we limit our experiments to be in simulation. However, we provide evidence indicating that the learned policies can be transferred to the real world in the future.

**A Surprising Finding**: While seemingly counterintuitive, we found that policies that have no access to shape information can manipulate a large number of previously unseen objects in all the three settings mentioned above. At the start of the project, we hypothesized that developing visual processing architecture for inferring shape while the robot manipulates the object would be the primary research challenge. On the contrary, our results show that it is possible to learn control strategies for general in-hand object re-orientation that are shape-agnostic. Our results, therefore, suggest that visual perception may be less important for in-hand manipulation than previously thought. Of course, we still believe that the performance of our system can be improved by incorporating shape information. However, our findings suggest a different framework of thinking: a lot can be achieved without vision, and that vision might be the icing on the cake instead of the cake itself.

## 2  Method

We found that *simple* extensions to existing techniques in robot learning can be used to construct a system for general object reorientation. First, to avoid explicit modeling of non-linear and frequent changes in the contact state between the object and the hand, we use model-free reinforcement learning (RL). An added advantage is that model-free RL is amenable to direct operation from raw point cloud observations, which is preferred for real-world deployment. We found that better policies can be trained faster using *privileged* state information such as the velocities of the object/fingertips that is easily available in the simulator but not in the real world. To demonstrate the possibility of transferring learned policies to the real world in the future, we overcome the need for privileged information using the idea of *teacher-student training* [14, 15]. In this framework, first, an expert or *teacher* policy ($\pi^{\mathcal{E}}$) is trained using privileged information. Next, the *teacher* policy guides the learning of a *student* policy ($\pi^{\mathcal{S}}$) that only uses sensory inputs available in the real world. Let the state space corresponding to $\pi^{\mathcal{E}}$ and $\pi^{\mathcal{S}}$ be $\mathbb{S}^{\mathcal{E}}$ and $\mathbb{S}^{\mathcal{S}}$ respectively. In general, $\mathbb{S}^{\mathcal{E}} \neq \mathbb{S}^{\mathcal{S}}$.

We first trained the teacher policy to reorient more than two thousand objects of diverse shapes (see Section 2.1). Next, we detail the method for distilling $\pi^{\mathcal{E}}$ to a student policy using a reduced state space consisting of only the joint positions of the hand, the object position, and the difference in orientation from the goal configuration (see Section 2.2.1). However, in the real world, even the object position and relative orientation must be inferred from sensory observation. Not only does this process require substantial engineering effort (e.g., a motion capture or a pose estimation system), but also inferring the pose of a symmetric object is prone to errors. This is because a symmetric object at multiple orientations looks exactly the same in sensory space such as RGBD images.

To mitigate these issues, we further distill $\pi^{\mathcal{E}}$ to operate directly from the point cloud and position of all the hand joints (see Section 2.2.2). We propose a simple modification that generalizes an existing 2D CNN architecture [16] to make this possible.

The procedure described above works well for manipulation with the hand facing upwards and downwards when a table is available as support. However, when the hand faces downward without an underlying support surface, we found it important to initialize the object in a stable configuration.

Finally, because gravity presents the primary challenge in learning policies with a downward-facing hand, we found that training in a curriculum where gravity is slowly introduced (i.e., *gravity curriculum*) substantially improves performance. These are discussed in Section 4.2

## 2.1 Learning the teacher policy

We use model-free RL to learn the teacher policy ($\pi^{\mathcal{E}}$) for reorienting an object ($\{O_i | i = 1, ..., N\}$) from an initial orientation $\alpha_0^o$ to a target orientation $\alpha^g$ ($\alpha_0^o \neq \alpha^g$). At every time step $t$, the agent observes the state $s_t$, executes the action $a_t$ sampled from the policy $\pi^{\mathcal{E}}$, and receives a reward $r_t$. $\pi^{\mathcal{E}}$ is optimized to maximize the expected discounted return: $\pi = \arg\max_{\pi^{\mathcal{E}}} \mathbb{E}\left[\sum_{t=0}^{T-1} \gamma^t r_t\right]$, where $\gamma$ is the discount factor. The task is successful if the angle difference $\Delta\theta$ between the object's current ($\alpha_t^o$) and the goal orientation ($\alpha^g$) is smaller than the threshold value $\bar{\theta}$, *i.e.*, $\Delta\theta \leq \bar{\theta}$.

To encourage the policy to be smooth, the previous action is appended to the inputs to the policy (*i.e.*, $a_t = \pi^{\mathcal{E}}(s_t, a_{t-1})$) and large actions are penalized in the reward function. We experiment with two architectures for $\pi^{\mathcal{E}}$: (1) an MLP policy $\pi_M$, (2) an RNN policy $\pi_R$. We use PPO [17] to optimize $\pi^{\mathcal{E}}$. More details about the training are in Section C.1 and Section C.2 in the appendix.

**Observation and action space**: We define $\mathbb{S}^{\mathcal{E}}$ to include joint, fingertip, object, and goal information as detailed in Table B.1 in the appendix. Note that $\mathbb{S}^{\mathcal{E}}$ does not include object shape or information about friction, damping, contact states between the fingers and the object, etc. We control the joint movements by commanding the relative change in the target joint angle ($q_t^{target}$) on each actuated joint (action $a_t \in \mathbb{R}^{20}$): $q_{t+1}^{target} = q_t^{target} + a_t \times \Delta t$, where $\Delta t$ is the control time step. We clamp the action command if necessary to make sure $|\Delta q_t^{target}| \leq 0.33$ rad. The control frequency is 60 Hz.

**Dynamics randomization**: Even though we do not test our policies on a real robot, we train and evaluate policies with domain randomization [18] to provide evidence that our work has the potential to be transferred to a real robotic system in the future. We randomize the object mass, friction coefficient, joint damping and add noise to the state observation $s_t$ and the action $a_t$. More details about domain randomization are provided in Table C.4 in the appendix.

## 2.2 Learning the student policy

We distill the teacher $\pi^{\mathcal{E}}$ into the student policy $\pi^{\mathcal{S}}$ using Dagger [19], a learning-from-demonstration method that overcomes the covariate shift problem. We optimize $\pi^{\mathcal{S}}$ by minimizing the KL-divergence between $\pi^{\mathcal{S}}$ and $\pi^{\mathcal{E}}$: $\pi^{\mathcal{S}} = \arg\min_{\pi^{\mathcal{S}}} D_{KL}\left(\pi^{\mathcal{E}}(s_t^{\mathcal{E}}, a_{t-1}) || \pi^{\mathcal{S}}(s_t^{\mathcal{S}}, a_{t-1})\right)$. Based on observation data available in real-world settings, we investigate two different choices of $\mathbb{S}^{\mathcal{S}}$.

### 2.2.1 Training student policy from low-dimensional state

In the first case, we consider a non-vision student policy $\pi^{\mathcal{S}}(s_t^{\mathcal{S}}, a_{t-1})$. $s_t^{\mathcal{S}} \in \mathbb{R}^{31}$ includes the joint positions $q_t \in \mathbb{R}^{24}$, object position $p_t^o \in \mathbb{R}^3$, quaternion difference between the object's current and target orientation $\beta_t \in \mathbb{R}^4$. In this case, $\mathbb{S}^{\mathcal{S}} \subset \mathbb{S}^{\mathcal{E}}$, and we assume the availability of object pose information, but do not require velocity information. We use the same MLP and RNN network architectures used for $\pi^{\mathcal{E}}$ on $\pi^{\mathcal{S}}$ except the input dimension changes as the state dimension is different.

### 2.2.2 Training student policy from vision

In the second case, $\mathbb{S}^{\mathcal{S}}$ only uses direct observations from RGBD cameras and the joint position ($q_t$) of the robotic hand. We convert the RGB and Depth data into a colored point cloud using a pinhole camera model [20]. Our vision policy takes as input the voxelized point cloud of the scene $W_t$, $q_t$, and previous action command $a_{t-1}$, and outputs the action $a_t$, *i.e.*, $a_t = \pi^{\mathcal{S}}(W_t, q_t, a_{t-1})$.

**Goal specification**: To avoid manually defining per-object coordinate frame for specifying the goal quaternion, we provide the goal to the policy as an object point cloud rotated to the desired orientation $W^g$, *i.e.*, we only show the policy how the object should look like in the end (see the top left of Figure C.5). The input to $\pi^{\mathcal{S}}$ is the point cloud $W_t = W_t^s \cup W^g$ where $W_t^s$ is the actual point cloud of the current scene obtained from the cameras. Details of obtaining $W_g$ are in Section C.2.

**Sparse3D-IMPALA-Net**: To convert a voxelized point cloud into a lower-dimensional feature representation, we use a sparse convolutional neural network. We extend the IMPALA policy architecture [16] for processing RGB images to process colored point cloud data using 3D convolution. Since

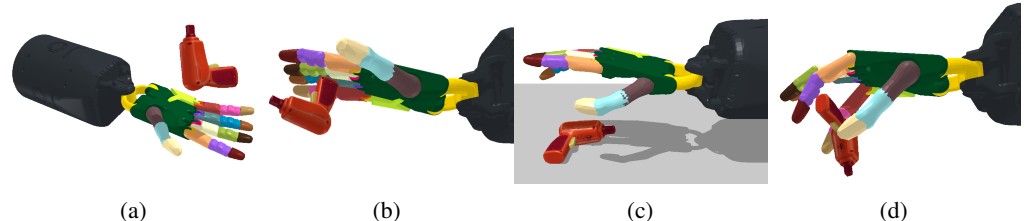

Figure 2: Examples of initial poses of the hand and object. **(a)**: hand faces upward. **(b)**, **(c)**, **(d)**: hand faces downward. **(b)**: both the hand and the object are initialized with random poses . **(c)**: there is a table below the hand. **(d)**: the hand and the object are initialized from the lifted poses.

many voxels are unoccupied, the use of regular 3D convolution substantially increases computation time. Hence, we use Minkowski Engine [21], a PyTorch library for sparse tensors, to design a 3D version of IMPALA-Net with sparse convolutions (*Sparse3D-IMPALA-Net*)[1]. The Sparse3D-IMPALA network takes as input the point cloud $W_t$, and outputs an embedding vector which is concatenated with the embedding vector of $(q_t, a_{t-1})$. Afterward, a recurrent network is used and outputs the action $a_t$. The detailed architecture is illustrated in Figure C.5 in the appendix.

**Mitigating the object symmetry issue**: $\pi^{\mathcal{E}}$ is trained with the the ground-truth state information $s_t^{\mathcal{E}}$ including the object orientation $\alpha_t^o$ and goal orientation $\alpha^g$. The vision policy does not take any orientation information as input. If an object is symmetric, the two different orientations of the object may correspond to the same point cloud observation. This makes it problematic to use the difference in orientation angles ($\Delta\theta \leq \bar{\theta}$) as the stopping and success criterion. To mitigate this issue, we use Chamfer distance [22] to compute the distance between the object point cloud in $\alpha_t^o$ and the goal point cloud (i.e., the object rotated by $\alpha^g$) as the evaluation criterion. The Chamfer distance is computed as $d_C = \sum_{a \in W_t^o} \min_{b \in W^g} \|a - b\|_2^2 + \sum_{b \in W^g} \min_{a \in W_t^o} \|a - b\|_2^2$, where $W_t^o$ is the object point cloud in its current orientation. Both $W_t^o$ and $W^g$ are scaled to fit in a unit sphere for computing $d_C$. We check Chamfer distance in each rollout step. If $d_C \leq \bar{d}_C$ ($\bar{d}_C$ is a threshold value for $d_C$), we consider the episode to be successful. Hence, the success criterion is $(\Delta\theta \leq \bar{\theta}) \vee (d_C \leq \bar{d}_C)$. In training, if the success criterion is satisfied, the episode is terminated and used for updating $\pi^{\mathcal{S}}$.

## 3  Experimental Setup

We use the simulated Shadow Hand [23] in NVIDIA Isaac Gym [24]. Shadow Hand is an anthropomorphic robotic hand with 24 degrees of freedom (DoF). We assume the base of the hand to be fixed. Twenty joints are actuated by agonist–antagonist tendons and the remaining four are under-actuated.

**Object datasets**: We use the EGAD dataset [25] and YCB dataset [26] that contain objects with diverse shapes (see Figure B.2) for in-hand manipulation experiments. EGAD contains 2282 geometrically diverse textureless object meshes, while the YCB dataset includes textured meshes for objects of daily life with different shapes and textures. We use the 78 YCB object models collected with the Google scanner. Since most YCB objects are too big for in-hand manipulation, we proportionally scale down the YCB meshes. To further increase the diversity of the datasets, we create 5 variants for each object mesh by randomly scaling the mesh. More details of the object datasets are in Section B.2.

**Setup for visual observations**: For the vision experiments, we trained policies for the scenario of hand facing upwards. We place two RGBD cameras above the hand (Figure B.4). The data from these cameras is combined to create a point cloud observation of the scene [20]. For downstream computation, the point cloud is voxelized with a resolution of $0.003\,\mathrm{m}$.

**Setup with the upward facing hand**: We first consider the case where the Shadow Hand faces upward and is required to reorient objects placed in the hand (see Figure 2a). We use the coordinate system where the $z$-axis points vertically upward and the $xy$-plane denotes the horizontal plane. The object pose is initialized with the following procedure: $xy$ position of the object's center of mass (COM) $p_{0,xy}^o$ is randomly sampled from a square region of size $0.09\,\mathrm{m} \times 0.09\,\mathrm{m}$. The center of this square is approximately located on the intersection of the middle finger and the palm so that the sampling region covers both the fingers and the palm. The $z$ position of the object is fixed to $0.13\,\mathrm{m}$ above the base of the hand to ensure that the object does not collide with the hand at initialization. The initial and goal orientations are randomly sampled from the full $SO(3)$ space.

---

[1]We also experimented with a 3D sparse convolutional network based on ResNet18, and found that 3D IMPALA-Net works better.

**Setup with the downward facing hand**: Next, we consider the cases where the hand faces downward. We experiment with two scenarios: with and without a table below the hand. In the first case, we place a table with the tabletop being $0.12\,\mathrm{m}$ below the hand base. We place objects in a random pose between the hand and the table so that the objects will fall onto the table. We will describe the setup for the second case (without a table) in Section 4.2.2.

**Evaluation criterion**: For non-vision experiments, a policy rollout is considered a success if $\theta \leq \bar{\theta}$. $\bar{\theta} = 0.1\,\mathrm{rad}$. For vision experiments, we also check $d_C \leq \bar{d}_C$ as another criterion and $\bar{\bar{\theta}} = 0.2\,\mathrm{rad}, \bar{d}_C = 0.01$. The initial and goal orientation are randomly sampled from $SO(3)$ space in all the experiments. We report performance as the percentage of successful episodes when the agent is tasked to reorient each training object 100 times from arbitrary start to goal orientation. We report the mean and standard deviation of success rate from 3 seeds.

# 4 Results

We evaluate the performance of reorientation policies with the hand facing upward and downward. Further we analyze the generalization of the learned policies to unseen object shapes.

## 4.1 Reorient objects with the hand facing upward

**Train a teacher policy with full-state information**   We train our teacher MLP and RNN policies using the full state information using all objects in the EGAD and YCB datasets separately. The progression of success rate during training is shown in Figure D.6 in Appendix D.1 . Figure D.6 also shows that using privileged information substantially speeds up policy learning. Results reported in Table 1 indicate that the RNN policies achieve a success rate greater than 90% on the EGAD dataset (entry B1) and greater than 80% on the YCB dataset (entry G1) without any explicit knowledge of the object shape[2]. This result is surprising because apriori one might believe that shape information is important for in-hand reorientation of diverse objects.

The visualization of policy rollout reveals that the agent employs a clever strategy that is invariant to object geometry for re-orienting objects. The agent throws the object in the air with a spin and catches it at the precise time when the object's orientation matches the goal orientation. Throwing the object with a spin is a dexterous skill that automatically emerges! One possibility for the emergence of this skill is that we used very light objects. This is not true because we trained with objects in the range of 50-150g which spans many hand-held objects used by humans (e.g., an egg weighs about 50g, a glass cup weighs around 100g, iPhone 12 weighs 162g, etc.). To further probe this concern, we evaluated zero-shot performance on objects weighing up to 500g[3] and found that the learned policy can successfully reorient them. We provide further analysis in the appendix showing that forces applied by the hand for such manipulation are realistic. While there is still room for the possibility that the learned policy is exploiting the simulator to reorient objects by throwing them in the air, our analysis so far indicates otherwise.

Next, to understand the failure modes, we collected one hundred unsuccessful trajectories on YCB dataset and manually analyzed them. The primary failure is in manipulating long, small, or thin objects, which accounts for $60\%$ of all errors. In such cases, either the object slips through the fingers and falls, or is hard to be manipulated when the objects land on the palm. Another cause of failures ($19\%$) is that objects are reoriented close to the goal orientation but not close enough to satisfy $\Delta\theta < \bar{\theta}$. Finally, the performance on YCB is lower than EGAD because objects in the YCB dataset are more diverse in their aspect ratios. Scaling these objects by constraining $l_{\max} \in [0.05, 0.12]$m (see Section 3) makes some of these objects either too small, too big, or too thin and consequently results in failure (see Figure D.7). A detailed object-wise quantitative analysis of performance is reported in appendix Figure D.10. Results confirm that sphere-like objects such as tennis balls and orange are easiest to reorient, while long/thin objects such as knives and forks are the hardest to manipulate.

**Train a student policy with a reduced state space**   The student policy state is $s_t^S \in \mathbb{R}^{31}$. In Table 1 (entries E1 and J1), we can see that $\pi_R^S$ can get similarly high success rates as $\pi_R^{\mathcal{E}}$. The

---

[2]More quantitative results on the MLP policies are available in Table D.5 in the appendix.

[3]We change the mass of the YCB objects to be in the range of $[0.3, 0.5]$kg, and test $\pi_R^{\mathcal{E}}$ from the YCB dataset on these new objects. The success rate is around $75\%$.

Table 1: Success rates (%) of policies tested on different dynamics distribution. $\bar{\theta} = 0.1\text{rad}$. DR: domain randomization and observation/action noise. X→Y: distill policy X into policy Y. The full table is in Table D.5.

| Exp. ID | Dataset | State | Policy | 1 Train without DR | 2 Train without DR | 3 Train with DR |
|---------|---------|-------|--------|--------------------|--------------------|------------------|
| | | | | Test without DR | Test with DR | Test with DR |
| B | EGAD | Full state | RNN | $95.95 \pm 0.8$ | $84.27 \pm 1.0$ | $88.04 \pm 0.6$ |
| E | | Reduced state | RNN→RNN | $91.96 \pm 1.5$ | $78.30 \pm 1.2$ | $80.29 \pm 0.9$ |
| G | YCB | Full state | RNN | $80.40 \pm 1.6$ | $65.16 \pm 1.0$ | $72.34 \pm 0.9$ |
| J | | Reduced state | RNN→RNN | $81.04 \pm 0.5$ | $64.93 \pm 0.2$ | $65.86 \pm 0.7$ |

last two columns in Table 1 also show that the policy is more robust to dynamics variations and observation/action noise after being trained with domain randomization.

## 4.2 Reorient objects with the hand facing downward

The results above demonstrate that when the hand faces upwards, RL can be used to train policies for reorienting a diverse set of geometrically different objects. A natural question to ask is, does this still hold true when the hand is flipped upside down? Intuitively, this task is much more challenging because the objects will immediately fall down without specific finger movements that stabilize the object. Because with the hand facing upwards, the object primarily rests on the palm, such specific finger movements are not required. Therefore, the hand facing downwards scenario presents a much harder exploration challenge. To verify this hypothesis, we trained a policy with the downward-facing hand, objects placed underneath the hand (see Figure 2b), and using the same reward function (Equation (1)) as before. Unsurprisingly, the success rate was $0\%$. The agent's failure can be attributed to policy needing to learn to both stabilize the object under the effect of gravity and simultaneously reorient it. Deploying this policy simply results in an object falling down, confirming the hard-exploration nature of this problem.

### 4.2.1 Reorient objects on a table

To tackle the hard problem of reorienting objects with the hand facing downward, we started with a simplified task setup that included a table under the hand (see Figure 2c). Table eases exploration by preventing the objects from falling. We train $\pi_M^{\mathcal{E}}$ using the same reward function Equation (1) on objects sampled from the EGAD and YCB datasets. The success rate using an MLP policy using full state information for EGAD and YCB is $95.31\% \pm 0.9\%$ and $81.59\% \pm 0.7\%$ respectively. Making use of external support for in-hand manipulation has been a challenging problem in robotics. Prior work approach this problem by building analytical models and constructing motion cones [27], which is challenging for objects with complex geometry. Our experiments show that model-free RL provides an effective alternative for learning manipulation strategies capable of using external support surfaces.

### 4.2.2 Reorient objects in air with hand facing downward

To enable the agent to operate in more general scenarios, we tackled the re-orientation problem with the hand facing downwards and without any external support. In this setup, one might hypothesize that object shape information (e.g., from vision) is critical because finding the strategy in Section 4.1 is not easy when the hand needs to overcome gravity and stabilize the object while reorienting it. We experimentally verify that even in this case, the policies achieve a reasonably high success rate without any knowledge of object shape.

**A good pose initialization is what you need**: The difficulty of directly training the RL policies when the hand faces downward is mainly because of the hard-exploration issue in learning to catch the objects that are moving downward. However, catching is not necessary for the reorientation. Even for human, we only reorient the object after we grasp it. More specifically, we first train an object-lifting policy to lift objects from the table, collect the ending state (joint positions $q_T$, object position $p_T^o$ and orientation $\alpha_T^o$) in each successful lifting rollout episode, and reset the hand and objects to these states (velocities are all 0) for the pose initialization in training the reorientation policy. The objects have randomly initialized poses and are dropped onto the table. We trained a separate RNN policy for each dataset using the reward function in Section C.2. The success rate on the EGAD dataset is $97.80\%$, while the success rate on the YCB dataset is $90.11\%$. Note that objects need to be grasped first to be lifted. Our high success rates on object lifting also indicate that using an anthropomorphic hand

Table 2: Performance of the student policy when the hand faces upward and downward

| Dataset | Upward | Downward (air) | Downward (air, $g$-curr) |
|---------|--------|----------------|--------------------------|
| EGAD | $91.96 \pm 1.5$ | $74.10 \pm 2.3$ | $/$ |
| YCB | $81.04 \pm 0.5$ | $45.22 \pm 2.1$ | $67.33 \pm 1.9$ |

Table 3: Zero-shot RNN policy transfer success rates (%) across datasets. **U.** (**D.**) means hand faces upward (downward). **FS** (**RS**) means using full-state (reduced-state) information.

| | EGAD $\rightarrow$ YCB | YCB $\rightarrow$ EGAD |
|-------|------------------------|------------------------|
| **U.FS** | $68.82 \pm 1.7$ | $96.41 \pm 1.2$ |
| **U.RS** | $59.64 \pm 1.8$ | $96.38 \pm 1.3$ |
| **D.FS** | $62.73 \pm 2.2$ | $85.45 \pm 2.9$ |
| **D.RS** | $55.30 \pm 1.3$ | $77.91 \pm 2.1$ |

Table 4: Vision policy success rate ($\bar{\theta} = 0.2\,\mathrm{rad}, \bar{d}_C = 0.01$)

| Object | Success rate (%) |
|--------|------------------|
| 025_mug | $89.67 \pm 1.2$ |
| 065-d_cups | $68.32 \pm 1.9$ |
| 072-b_toy_airplane | $84.52 \pm 1.4$ |
| 073-a_lego_duplo | $58.16 \pm 3.1$ |
| 073-c_lego_duplo | $50.21 \pm 3.7$ |
| 073-e_lego_duplo | $66.57 \pm 3.1$ |

makes object grasping an easy task, while many prior works [28, 29] require much more involved training techniques to learn grasping skills with parallel-jaw grippers. After we train the lifting policy, we collect about 250 ending states for each object respectively from the successful lifting episodes. In every reset during the reorientation policy training, ending states are randomly sampled and used as the initial pose of the fingers and objects. With a good pose initialization, policies are able to learn to reorient objects with high success rates. $\pi_R^{\mathcal{E}}$ trained on EGAD dataset gets a success rate more than $80\%$ while $\pi_R^{\mathcal{E}}$ trained on YCB dataset gets a success rate greater than $50\%$. More results on the different policies with and without domain randomization are available in Table D.6 in the appendix. This setup is challenging because if at any time step in an episode the fingers take a bad action, the object will fall.

**Improving performance using gravity curriculum**: Since the difficulty of training the reorientation policy with the hand facing downward is due to the gravity, we propose to build a gravity curriculum to learn the policy $\pi^{\mathcal{E}}$. Since $\pi^{\mathcal{E}}$ already performs very well on EGAD objects, we apply gravity curriculum to train $\pi^{\mathcal{E}}$ on YCB objects. Our gravity curriculum is constructed as follows: we start the training with $g = 1\,\mathrm{m/s^2}$, then we gradually decrease $g$ in a step-wise fashion if the evaluation success rate ($w$) is above a threshold value ($\bar{w}$) until $g = -9.81\,\mathrm{m/s^2}$. More details about gravity curriculum are in Section C.4. In Table D.6 (Exp Q and T) in the appendix, we can see that adding gravity curriculum ($g$-curr) significantly boost the success rates on the YCB dataset.

### 4.3 Zero-shot policy transfer across datasets

We have shown the testing performance on the same training dataset so far. How would the policies work on a different dataset? To answer this, we test our policies across datasets: policies trained with EGAD objects are now tested with YCB objects and vice versa. We used the RNN policies trained with full-state information and reduced-state information respectively (without domain randomization) and tested them on the other dataset with the hand facing upward and downward. In the case of the hand facing downward, we tested the RNN policy trained with gravity curriculum. Table 3 shows that policies still perform well on the untrained dataset.

### 4.4 Object Reorientation with RGBD sensors

In this section, we investigate whether we can train a vision policy to reorient objects with the hand facing upward. As vision-based experiments require much more compute resources, we train one vision policy for each object individually on six objects shown in Table 4. We leave training a single vision policy for all objects to future work. We use the expert MLP policy trained in Section 4.1 to supervise the vision policy. We also performed data augmentation on the point cloud input to the policy network at each time step in both training and testing. The data augmentation includes the random translation of the point cloud, random noise on the point positions, random dropout on the points, and random variations on the point color. More details about the data augmentation are in Section D.5. We can see from Table 4 that reorienting the non-symmetric objects including the toy and the mug has high success rates (greater than $80\%$). While training the policy for symmetric objects is much harder, Table 4 shows that using $d_C$ as an episode termination criterion enables the policies to achieve a success rate greater than $50\%$.

## 5 Related Work

Dexterous manipulation has been studied for decades, dating back to [30, 31]. In contrast to parallel-jaw grasping, pushing, pivoting [32], or pick-and-place, dexterous manipulation typically involves continuously controlling force to the object through the fingertips of a robotic hand [33]. Some prior works used analytical kinematics and dynamics models of the hand and object, and used trajectory optimization to output control policies [1, 2, 34] or employed kinodynamic planning to find a feasible motion plan [35]. However, due to the large number of active contacts on the hand and the objects, model simplifications such as simple finger and object geometries are usually necessary to make the optimization or planning tractable. Sundaralingam and Hermans [34] moved objects in hand but assumes that there is no contact breaking or making between the fingers and the object. Furukawa et al. [36] achieved a high-speed dynamic regrasping motion on a cylinder using a high-speed robotic hand and a high-speed vision system. Prior works have also explored the use of a vision system for manipulating an object to track a planned path [37], detecting manipulation modes [38], precision manipulation [39] with a limited number of objects with simple shapes using a two-fingered gripper. Recent works have explored the application of reinforcement learning to dexterous manipulation. Model-based RL works learned a linear [4, 7] or deep neural network [5] dynamics model from the rollout data, and used online optimal control to rotate a pen or Baoding balls on a Shadow hand. However, when the system is unstable, collecting informative trajectories for training a good dynamics model that generalizes to different objects remains challenging. Another line of works uses model-free RL algorithms to learn a dexterous manipulation policy. For example, OpenAI et al. [11] and OpenAI et al. [12] learned a controller to reorient a block or a Rubik's cube. Van Hoof et al. [40] learned the tactile informed policy via RL for a three-finger manipulator to move an object on the table. To reduce the sample complexity of model-free learning, [9, 13, 8, 41, 6] combined imitation learning with RL to learn to reorient a pen, open a door, assemble LEGO blocks, etc. However, collecting expert demonstration data from humans is expensive, time-consuming, and even incredibly difficult for contact-rick tasks [8]. Our method belongs to the category of model-free learning. We use the teacher-student learning paradigm to speed up the deployment policy learning. Our learned policies also generalize to new shapes and show strong zero-shot transfer performance. To the best of our knowledge, our system is the first work that demonstrates the capabilities of reorienting a diverse set of objects that have complex geometries with both the hand facing upward and downward. A recent work [42] (after our CoRL submission) learns a shape-conditioned policy to reorient objects around $z$-axis with an upward-facing hand. Our work tackles more general tasks (more diverse objects, any goal orientation in $SO(3)$, hand faces upward and downward) and shows that even without knowing any object shape information, the policies can get surprisingly high success rates in these tasks.

## 6 Discussion and Conclusion

Our results show that model-free RL with simple deep learning architectures can be used to train policies to reorient a large set of geometrically diverse objects. Further, for learning with the hand facing downwards, we found that a good pose initialization obtained from a lifting policy was necessary, and the gravity curriculum substantially improved performance. The agent also learns to use an external surface (i.e., the table). The most surprising observation is that information about shape is not required despite the fact that we train a single policy to manipulate multiple objects. Perhaps in hindsight, it is not as surprising – after all, humans can close their eyes and easily manipulate novel objects into a specific orientation. Our work can serve a strong baseline for future in-hand object reorientation works that incorporate object shape in the observation space.

While we only present results in simulation, we also provide evidence that our policies can be transferred to the real world. The experiments with domain randomization indicate that learned policies can work with noisy inputs. Analysis of peak torques during manipulation (see Figure D.12 in the appendix) also reveals that generated torque commands are feasible to actuate on an actual robotic hand.

Finally, Figure D.10 and Figure D.11 in the appendix show that the success rate varies substantially with object shape. This suggests that in the future, a training curriculum based on object shapes can improve performance. Another future work is to directly train one vision policy for a diverse set of objects. A major bottleneck in vision-based experiments is the demand for much larger GPU memory. Learning visual representations of point cloud data that can ease the computational bottleneck is, therefore, an important avenue for future research.

**Acknowledgments**

We thank the anonymous reviewers for their helpful comments in revising the paper. We thank the members of Improbable AI lab for providing valuable feedback on research idea formulation and manuscript. This research is funded by Toyota Research Institute, Amazon Research Award, and DARPA Machine Common Sense Program. We also acknowledge the MIT SuperCloud and Lincoln Laboratory Supercomputing Center for providing HPC resources that have contributed to the research results reported within this paper.

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
