# OpenReview forum: "A System for General In-Hand Object Re-Orientation"
_robot-learning.org/CoRL/2021/Conference — CoRL2021 Oral_

### Official Review · Reviewer_uUVE · 2021-07-23

**Originality:** Very Good
**Technical Quality:** Excellent
**Clarity Of Presentation:** Excellent
**Impact:** 4

**Recommendation:**

Strong Accept: I recommend accepting the paper and will argue for my recommendation even if other reviewers hold a different opinion.

**Summary:**

This paper presents a system in simulation for in-hand reorientation of novel objects. The work leverage privileged information from the simulator to train a model-free teacher policy and then applies Dagger to learn a model-free student policy which has access to only sensor inputs that are generally available in the real world.


**Issues:**

W2
- Besides the domain randomization tried, I believe there will likely be more considerations needed for zero shot sim2real to be possible -- this is just based on the fact that while simulation photo-realism has come a long way existing simulators still massively struggle (and make strong assumptions) when it comes to simulating contact. One could sacrifice sim speed for fidelity, but that would likely not serve well for a model-free RL pipeline.
- The simulation/task seem to get cut off when the success condition is triggered, but it seems that some of the executions were mid motion. This in practice would lead to overshooting goal, instability, dropping object, etc. A fix to consider would be to add some zero velocity criteria (so object is stable in-hand) on top of the current goal reward.
- How do we scale model-free RL to real world and lead to more 'natural' motions?
    - An observation from the experiment videos is that there tends to be a lot of flaying and extra finger motions, likely since the reward is not capturing these.
    - What about ego centric perception and thus heavy occlusion when working in the real-world.
    - How do we continue to learn online in the real world since we won't have access to privileged information anymore.
The above questions are obviously hard and open research problems, but some discussion in the context of the limitations of this paper would be beneficial.

**Reviewer Expertise:**

Very good: Comprehensive knowledge of the area

**Strengths And Weaknesses:**

Strengths
+ I consider this a 'systems' paper as it builds on mostly known algorithmic advances but effectively puts them together in a 'simple' manner to demonstration fairly strong results on a hard in-hand reorientation task.
+ The experiments are quite comprehensive and the paper makes good faith effort in arguing the applicability of the method to a real setting despite the results demonstrated in sim only.
+ Results are impressive and will serve the community well as a strong starting baseline for future work.

Weaknesses
- w1: I expect/encourage the authors to release code, models, and datasets otherwise much of the value of the contribution would be lost.
- W2: Insights gained from experiments were quite helpful, it would be good to have a more extended discussion on the limitations.

**Summary Of Recommendation:**

Overall, this is a solid paper that makes advancements in in-hand manipulation.

---

> ### Author Response · Authors · 2021-08-30
> **Response to Reviewer uUVE**
>
> We would like to thank reviewer uUVE for the helpful comments. We address the reviewer's concerns as follows:
>
> **Q: "I expect/encourage the authors to release code, models, and datasets otherwise much of the value of the contribution would be lost."**
>
> **A:** Yes, we will release the code that can reproduce the results in the paper upon the paper's acceptance.
>
> **Q: A more extended discussion on the limitations. Also, How do we scale model-free RL to real world and lead to more 'natural' motions?**
>
> **A:** Due to the page limit, we have provided as much discussion as possible in the submission. We are happy to add more discussions here.
>
> * Currently, due to the compute limit, we are only able to train a separate vision policy for each object even though we can train one single state-based policy for all objects. Future work should look at efficiently train a single vision policy that can reorient many objects.
>
> * Figure D.10 and D.11 show that the success rates for objects with different shapes vary significantly. Future work should investigate methods for improving the policy's performance on objects that are not working as well.
>
> * While we have demonstrated surprisingly good performance of the learned policies without using any object shape information, it is an important future work to see how we can incorporate shape into the policy to improve performance. It is possible that this would improve performance on objects our current method suffers on.
>
> * As the reviewer mentioned, sometimes some fingers can move unnaturally during the reorientation. How to make the hand reorient object more human-like is an interesting direction. One possibility is to modify the reward function to include energy cost, coupling between fingers, etc. Another way is to make use of human hand demonstrations.
>
> * If the camera can only be placed in an ego-centric viewpoint, as mentioned by the reviewer, the occlusions will be more severe.  One way to mitigate occlusion is to augment the policy with a memory of past time steps or memory of past manipulation of the same object or one of a similar shape. Another way is adding more cameras as is commonly done in many mobile robot platforms and in our work.
>
> * Continuously learning in the real world is challenging. It requires reward computation, resets, etc. Future work investigating reset-free learning, use of self-supervised learning in conjunction with RL are likely to benefit. Another challenge is in robust hardware that is capable of running for millions of iterations. Another avenue is to incorporate human demonstrations to aid learning of manipulation policies.
>
> **Q: Episodes terminate after the goal orientation is reached. There could be overshooting goal.**
>
> **A:** Reviewer UVfU also asked this question. Our common response to both reviewers is below:
>
> Since we randomize the initial and goal orientations during training and generously reward the agent for manipulating the object into the goal orientation, we hypothesize that the goal acts as an "attractor" at policy's convergence. Therefore, during testing, even if the policy overshoots, it knows how to reorient the object back to the goal orientation. In such a scenario, if the policy overshoots too much, there will be oscillations, and the object will be reoriented back-and-forth around the goal orientation. Empirically, we observe that the policy doesn't show such oscillations as can be seen in 16 new videos on the project website ([https://sites.google.com/view/in-hand-reorientation](https://sites.google.com/view/in-hand-reorientation)) where the goal orientation changes after the previous goal orientation is met ("Continuing Reorientation"). Additionally, we have included 16 new videos where the episode is not terminated after the goal orientation is reached ("No Immediate Termination after Reaching the Goal Orientation"). We can see from these videos that even though the objects are not "still" on reaching the goal orientation, there is no significant overshooting and the objects are largely stable.
>
> With that being said, there is certainly room to improve stability at the goal state. For example, during training, we can set the success criterion to be that the object needs to stay close to the goal orientation for several consecutive time steps or the velocity of the object needs to be 0, etc. These small improvements can be easily incorporated into our work. However, since our setup already works well. We leave it to future work as rerunning all the experiments with such modifications will take more time than the rebuttal period.

---

> > ### Comment · Reviewer_uUVE · 2021-09-02
> > **response to authors**
> >
> > Thank you for the extra discussion and the additional videos. Having read the other reviews and rebuttal I will keep my score and support accepting the paper.

---

### Official Review · Reviewer_Cfuq · 2021-07-26

**Originality:** Very Good
**Technical Quality:** Good
**Clarity Of Presentation:** Fair
**Impact:** 4

**Recommendation:**

Strong Accept: I recommend accepting the paper and will argue for my recommendation even if other reviewers hold a different opinion.

**Summary:**

This paper includes two main contributions. First, the authors developed an approach to transfer (using DAGGER) a teacher policy trained (using model-free RL) with full state information to a student policy trained with only sensory information. To achieve this, they exploited the fact that in simulations the state information and the correlated sensory information are simultaneously available, which enables the student to imitate the teacher’s action and learn a policy that maps sensory information to the same actions. They applied this design to in-hand reorientation of objects and demonstrated that their approach allows learning of successful reorientation for a much larger set of object shapes than has been attempted by the research community. Second, the authors also developed an effective way of training the agent to do in-hand reorientation with the robotic hand facing downwards. To achieve this, the authors had to develop both a lift policy that is used to identify good initial poses and a “gravity” curriculum where the gravitational acceleration is changed gradually from 1.0 to -9.8.

**Issues:**

a.	Title: “A Simple Method for Complex In-Hand Manipulation”. From the text, it is not clear what the “method” is. Is it the distillation of policies trained with state information into ones that do not use state information? If so, how do the various arrangements needed for learning manipulation with hand facing down fit into the method? Also not clear is what the authors mean by “Simple”. In what sense is the method simple?

b.	Regarding the fact that this paper includes two projects: (1) transfer and (2) in-hand reorientation with hand facing down, the authors made no attempt to link the two themes. For (2), the authors ended up developing a lot of machinery (lifting policy, gravity curriculum, table support etc) to investigate and to get it to work. Doesn’t all these under (2) make “the method” very complex? Or if (1) and (2) are really different, maybe we should do two separate papers? If (1) and (2) are related, how should or could the transfer method be adapted in a curriculum context?

c.	In the eyes of this reviewer, the authors may have missed a great opportunity to exploit a hand-orientation curriculum. That is, instead of artificially tweaking gravity, we could much more naturally make hand orientation change according to a curriculum. (See o. and r. below for more.) This could allow the authors to eliminate most of the complexity noted above in b. above. Moreover, there might also be an opportunity of linking the two parts of the paper together, if a hand-orientation curriculum could be organized with the curriculum for the teacher requiring fine-grained sampling of orientations but that for the student requiring much more coarse-grained sampling of orientations (or something like that).

d.	Figure 1: presumably misplaced above the Abstract.

e.	Line 3 in the Abstract: “a simple model-free framework” -- the whole abstract contains zero information about either the method (as in the title) or the framework. Nor does it give any sense to the word “Simple” in the title and here. Presumably, the unique or novel features of the method or the framework should be explained or at least noted in the Abstract as well as how such features make the method or framework simple.

f.	Lines 18-19: “These two tasks are harder because the object is in an intrinsically unstable configuration”. This does not seem to be true of the case where the object being manipulated is supported by a table.

g.	Line 40: Among the desiderata, (d) could be more clearly formulated in explicit contrast with approaches that require accurate state measurement (e.g. through motion capture).

h.	Line 43: “building a simple framework”, again no real information was given as to why and how the framework is simple in this Introduction.

i.	Line 63-66: This sentence adds nothing. It seems to be a wasted sentence because it started to talk about how the student policy is trained and then deferred to Section 2.2.1 immediately. It is also not clear how this sentence connects with the next one that starts with “However, …”

j.	Line 94: should introduce delta q_t^target explicitly.

k.	Section 2.2: it is probably going to help a lot if a table is included to directly compare the observations available to the teacher (Full State) and those available to the student (Reduced State).

l.	Figure 2: gap is needed between the main text and the caption for Figure 2.

m.	Lines 189-190: theta^bar under the two settings are respectively 0.1 and 0.2. Any principled reason why? Also, d^bar_C is set to 0.01. How does that combine with theta^bar = 0.2?

n.	Line 192: “reorients” => “reorient”.

o.	Section 42 Lines 235-245: The discussion in this paragraph is important but seems more narrow than warranted. First, the authors state “A natural question to ask is then does this still hold true when the hand is flipped upside down.” (Lines 236-237) But a more natural question seems to be if the hand’s orientation deviates from facing upwards at all. There are many possible orientations the hand could take. Facing up and facing down are just two extremes. In fact, the gravity curriculum proposed by the authors could be possibly replaced by a hand orientation curriculum that could be much more natural and potentially enable learning of policy structures much more generalizable. Second, “Unsurprisingly, the agent is unable to learn …” is possibly only unsurprising if the hand is putting in the extreme orientation of facing downward. The agent has no opportunity of learning effect of gravity for different orientations, with the two studied in the paper being only two extremes occupying a whole continuum of rich structures.

p.	Section 4.2.1: some comment/information about the strategy learned by the agent is expected. For example, does the agent leave the object to be supported by the table alone without any contact between maneuvers? If so, how that may have simplified the learning for the agent? If not, how does the support help? The reader is left wondering …

q.	Lines 267-270: the method of training the agent to learn a lifting policy and then use it to determine initial pose, while effective, seems to add a lot of complexity to the method. The reader is left to wonder why the method/framework is still characterized as “simple”.

r.	Line 287: “we start the training with g = 1 m / s^2,” -- This means there is (anti-)gravitational force moving the object upward into the hand. Why is this needed? How could this be justified?

s.	Line 309: “high success rate” -- there is no reference / baseline / comparison offered to support the claim that the success rate is high.

t.	Line 351: “this policy” => “the teacher policy” might be more clear.


**Reviewer Expertise:**

Very good: Comprehensive knowledge of the area

**Strengths And Weaknesses:**

Strengths

i.	The transfer approach developed by the authors is a promising approach. It is capable of leveraging increasingly better simulators and increasingly bigger datasets or model sets of diverse physical objects. The fact that the authors could get reorientation to work for a much larger set of object shapes than before could be believably attributed to the effectiveness of their approach.

ii.	Getting reorientation to work well enough with the robotic hand facing down is not trivial. The authors did a fairly thorough study of this highly specific but very challenging case.

iii.	The empirical results reported are largely believable.

iv.	The system integration, while not involving physical robots, is nontrivial.

Weaknesses

i.	There seems to be two papers here: the first one about the transfer method and the second one about learning reorientation when the robotic hand is facing downward. Either of these have enough content and importance. What is unfortunately is that the authors completely failed to relate these two aspects of the paper/project in an organic way. See Issues.a below for more discussion.

ii.	It should be noted that the effectiveness of the transfer method is not fully validated in the sense of supporting a full transfer to physical robot. (Of course getting Shadow hand to work physically with deep RL is in itself a huge challenge.) The reason this ultimate transfer to the real world is important is because it will test how heavily the transfer depends on the full coincidence of state information and sensory information that is likely only obtainable in simulation. The method's strength won’t be adequately established until evaluated against a full transfer to the real world. Similarly, the method's alleged “simplicity” (which the authors never made adequately explicit, but presumably is another way of calling the straightforward exploitation of the coincidence of state and sensory info in simulation) may become a liability if transfer to the real world cannot be made to work well.

iii.	The draft in its current state is quite rough with various writing and formatting issues (pointed out in Issues below). But the presentation overall suffers from a degree of being scattered. In addition to the major issue of there being two projects/papers in one, some other examples include specifics and motivations for variations such as MLP vs. RNN being missing.


**Summary Of Recommendation:**

In spite of the weaknesses identified earlier and the numerous issues identified below, this reviewer believes that the authors have done enough and achieved enough to warrant attention from and discussion by the broader CoRL community. The presentation in the paper, which needs to be significantly improved in the final submission, does manage to get the main contribution across as compelling research. Either of the two major parts of the paper could stimulate and engender much meaningful research conversations.

---

> ### Author Response · Authors · 2021-08-30
> **Response to Reviewer Cfuq (1)**
>
> We sincerely thank reviewer Cfuq for the helpful feedback and suggestions on our paper. We address the reviewer's concerns below.
>
> **Q: "There seems to be two papers here: the first one about the transfer method and the second one about learning reorientation when the robotic hand is facing downward." "For (2), the authors ended up developing a lot of machinery (lifting policy, gravity curriculum, table support etc) to investigate and to get it to work. "**
>
> **A:** We are sorry that the two aspects of our work came out as disconnected. Our overall goal is to perform in-hand re-orientation of objects in general scenarios. These scenarios involve the hand facing upwards, downwards, and in some cases using an external surface. The teacher-student learning is applicable to all these scenarios. We employ this paradigm because it is easier and faster to learn teacher policies with privileged state information only available in the simulator (as shown in the rightmost figure in Figure D.6). Next, because we want these policies to be useful for real-world manipulation eventually, we distill this policy (i.e., student policy) to work only with sensory data available in the real world.
>
> The second set of machinery — the lifting policy and the gravity curriculum were used for manipulation with hand-facing downwards. The gravity curriculum is used to improve the teacher policy's performance. The student-teacher paradigm is complementary to these innovations and is necessary for superior performance on the student policy. We, therefore, feel that both the teacher-student learning and the machinery for downward-facing manipulation is necessary for the hand to generally re-orient objects.
>
> **Q: What does "Simple" refer to?**
>
> **A:** We acknowledge that the term "simple" has been used subjectively, and we never defined what is simple. We call our method simple because (i) we do not make use of explicit models of the object, the manipulator, or the contact dynamics. Our method builds upon a well-established model-free RL algorithm, PPO. The three key components: teacher-student learning, gravity curriculum, and stable initialization of objects are general ideas that are likely to be useful even outside the context of in-hand re-orientation tasks. (ii) Secondly, we don't require any special pre-processing of sensory observations. E.g., our vision-based policies can work directly from RGBD point clouds captured by cameras. We will add this clarification in the text. Also, we are not tied to the use of the word "simple". If the reviewer strongly feels we should not use the word "simple", we are happy to change the title.
>
> **Q: " the effectiveness of the transfer method is not fully validated in the sense of supporting a full transfer to physical robot"**
>
> **A:** We have not evaluated the policies on the physical robot due to limited lab access in the pandemic. We agree with the reviewer that the true test is real-world deployment. We could have presented only part of of our machinery in the paper, the method required for downward-facing object re-orientation. However, if this machinery used privileged observation only available in simulation, then there would be no chance of transfer to the real world. To ensure that there is a realistic chance for such transfer we presented the student-teacher paradigm that enables us to train policies with realistic sensory observations. Further, we provide a detailed analysis of how the performance varies due to noise in sensing and uncertainty in simulation parameters. We will better connect the two parts of the paper in the revised manuscript.
>
> **Q: Motivations for variations such as MLP vs. RNN are missing**
>
> **A:** We thank the reviewer for pointing this out. We have added the motivation in the Appendix Section D1 in the updated manuscript. The motivation is:
>
> - We hypothesized by temporal integration RNN teacher policies can implicitly represent object shape information and might perform better. However, we experimentally found that this is not the case and the MLP policy performs equally well (see Table 1).
> - For the student policies, RNN outperforms MLP policies because student policies do not have access to privileged state information, and accumulating information over time improves performance.

---

> > ### Author Response · Authors · 2021-08-30
> > **Response to Reviewer Cfuq (2)**
> >
> > **Q: "the authors may have missed a great opportunity to exploit a hand-orientation curriculum"**
> >
> > **A:** Thanks for the suggestion. Firstly, we want to point out that as shown in the gravity curriculum is not "critical" for manipulation (Exp. ID S in Table D.5), but it improves performance substantially (Exp. ID T in Table D.5).
> >
> > Secondly, we would like to draw parallel between the gravity and the hand-curriculum. Theoretically, instead of changing the orientation of the hand, we can change the direction of the gravity vector. Because all that matters is the relative relationship between gravity and hand orientation, it will have the same effect. In our work, we experimented with a limited gravity curriculum where the direction was fixed, but we changed the magnitude. We agree that it would indeed be an exciting avenue of future work to explore the hand-orientation curriculum/gravity direction curriculum to expand the scope of what the learned policies can achieve.
> >
> > Thirdly, we want to point out that the hand curriculum does not necessarily simplify the setup. We would still need a good object pose initialization and therefore the lifting policy. Please note that the use of a table is not part of the machinery for downward-facing manipulation. It is a separate experiment to probe can the learned policies make use of an external surface if it is available. Lastly, we will have to come up with a mechanism that decides when to change the orientation of the hand/gravity, which may or may not be simple.
> >
> > Therefore, we believe that hand-orientation curriculum is not a replacement for our machinery, but is complementary to our work. It will push the span of in-hand re-orientation further and will be an exciting avenue for future work. Thanks once again for the valuable suggestion.
> >
> > **Q:  Why is $g$ initialized as  $+1m/s^2$ (an (anti-)gravitational force)?**
> >
> > **A:** $+1m/s^2$ gives a small anti-gravitational force and this force can help push the object towards the hand (which makes it similar to the case of the hand facing upward). This makes learning easier. One can also start $g$ with $0m/s^2$, we found that setting $g_0$ to be $+1m/s^2$ marginally speeds up the learning.
> >
> > **Q: The abstract contains zero information about either the method (as in the title) or the framework. Line 43: “building a simple framework”, again no real information was given as to why and how the framework is simple in this Introduction.**
> >
> > **A:** Thanks for pointing it out. We will revise the draft to specify what simple means (which we have answered in the second question above).
> >
> > **Q: Lines 189-190: theta^bar under the two settings are respectively 0.1 and 0.2. Any principled reason why? Also, d^bar_C is set to 0.01. How does that combine with theta^bar = 0.2?**
> >
> > **A:** The two settings referred by the reviewer are training the student vision policy and training the policies (either teacher or student policies) with the full state information. Vision policy can be less accurate than the full-state policy as it does not have direct access to the object orientation and information loss resulting from point cloud voxelization. Therefore, we used a slightly higher threshold for the vision's policy reorientation error. However, to allow direct comparison between state and vision-based policies we will include the performance of vision policy with the threshold set to 0.1rad in the final version of our paper. At the same time please note that 0.2rad is still a small threshold (around 11 degrees).
> >
> > The combination of $\theta$ and $d_C$ is explained in Section 2.2.2 (Mitigating the object symmetry issue). Specifically when ($\Delta\theta\leq\bar{\theta}$) or ($d_C\leq \bar{d}_C$) is true, we declare success.
> >
> > **Q: Section 4.2.1: some comment/information about the strategy learned by the agent is expected.**
> >
> > **A:** As we can see from the videos on the project website, the hand reorients the objects in different ways depending on their shape and size. One way it learned is that it reorients the object while having the object always touching the table (sometimes it looks like a rolling motion) until the object is in the desired orientation. And at the end of the orientation, the hand might lift the object up to reach the final orientation. Another way it learned is that it grasps the object, reorients it a bit, puts it down on the table, and repeatedly performs grasping and placing the object on the table until the orientation matches with the target orientation.

---

> > > ### Comment · Reviewer_Cfuq · 2021-09-04
> > > **Response to Rebuttal**
> > >
> > > I would like to thank the authors for their detailed and careful rebuttal. I also find the addition of the videos for the "No immediate termination" case (in response to reviewer UVfU) to be compelling. I feel the authors have done enough and clarified enough to warrant a Strong Accept and have thus updated my recommendation accordingly.
> > >
> > > As to "simple", I remain unconvinced that that is the most helpful characterization. But may the authors really like the contradistinction with "complex". I won't object to that, but am still wondering ... ;-)

---

### Official Review · Reviewer_UVfU · 2021-07-26

**Originality:** Good
**Technical Quality:** Very Good
**Clarity Of Presentation:** Good
**Impact:** 3

**Recommendation:**

Strong Accept: I recommend accepting the paper and will argue for my recommendation even if other reviewers hold a different opinion.

**Summary:**

This paper presents a method of learning to manipulate a wide range of objects above and below the palm of the dexterous shadow-hand. Their method involves training a teacher policy with privileged information from the simulator (such as the true ground-truth state information), and then using it to distill a policy with Dagger into a student policy that learns to take actions in joint space directly from joint positions and a RGBD point cloud of the object. They compare both MLP and RNN architectures, and train and test with and without domain  randomization, on both the YCB and EGAD object datasets. They also make architectural contributions in their visual policy architecture by designing a sparse 3D convolutional version of the IMPALA architecture, which takes raw point clouds as input and outputs actions.

**Issues:**

- The tables are a bit difficult to quickly scan for which method shows the best performance, bolding the best performing method might be informative in drawing conclusions later from the results.
- There are multiple instances in the text where there are object variations in the scale, mass, and aspect ratios of the objects that are scattered throughout the sections (3, and 4.1). It would be helpful if the success rates for each of these different variations were compiled in a table in the appendix (C or D), so that their effects relative to one another on the performance of the method could be communicated more clearly.
- Table D.5 shows the performance of their vision model trained and tested with random transformations done on the point cloud. It would be useful to show the effect of these perturbations with an ablation study showing the effect that each of these transformations (and all of them combiend) on a model that is trained without them.
- I was unable to find the experimental results on the zero-shot training performance on objects weighing up to 500g referenced on line 212 in the paper and appendix.

**Reviewer Expertise:**

Very good: Comprehensive knowledge of the area

**Strengths And Weaknesses:**

Strengths:

- The sheer scope and variation across objects tested with this method, and the range of different policy architectures and approaches tested makes this paper extremely thorough in its analysis of this reorientation task.
- The authors deliberately choose joint position control with RGBD sensors to make their method easier to transfer to a real robot environment.
- The use of the gravity curriculum to successfully train the hand to manipulate the object upside-down is also a considerable feat and seems to work well in simulation. It remains to be seen if this policy is transferrable to a real environment.
- Their method also shows the ability to zero-shot transfer across different object datasets (with YCB -> EGAD showing better transfer results)

Weaknesses:

- The authors mention that their current training strategy terminates the episode if the time limit is reached, or the hand is successful at reposing the object to the goal orientation. However, this does not guarantee that the final state of the object is stably grasped by the hand. This is apparent given one of the learned strategies for the upward-facing hand throws the object up with a spin to get it to the correct pose. While it is an interesting emergent skill, it is a limitation that their method does not guarantee that the object can be used by the hand after being reposed.
- The authors claim that position control is preferred to torque control for sim-to-real transfer. However since their method is not actually tested on a real robot, and they do domain randomization to be invariant to joint and tension dampening, the stated reasoning is not fully consistent.
- The student network operates with an RGBD point cloud and position of all the hand joints. Point clouds are in general noisy and might contain even more noise than tested in simulation with domain randomization when factoring the occlusions and interaction between the hand and the object. The transferability between sim and real of this policy remains a considerable challenge.
- The notable issue of reorienting symmetric objects with RGBD sensors discussed in the paper makes their method less robust for certain kinds of objects. It might be worth exploring whether using object keypoints can help reduce some of these effects.

**Summary Of Recommendation:**

I would like to recommend this paper purely on its merits for having extensively explored how to enable dexterous reposing of objects with model-free reinforcement learning methods using 3D Convolutions and teacher-student training. While they do not show their method working on a real hardware platform, due to the current state of the pandemic, they do extensively test their method with different architectures and object datasets. Additionally, they demonstrate the generality of their approach by showing zero-shot policy transfer across these datasets. One limitation of their method include the fact that the environment terminates episodes after successfully reorienting the object to the goal pose without seeing if it falls/can hold that position after the episode ends.

---

> ### Author Response · Authors · 2021-08-30
> **Response to Reviewer UVfU (1)**
>
> We sincerely thank reviewer UVfU for the feedback on our paper. We address the reviewer's concerns as follows.
>
> **Q: Terminating the episode on success does not guarantee that the final state of the object is stably grasped by the hand.**
>
> **A**: We empirically found that in most cases the final states are stable in the successful episodes. Since we randomize the initial and goal orientations during training and generously reward the agent for manipulating the object into the goal orientation, we hypothesize that the goal acts as an "attractor" at policy's convergence. Therefore, during testing, even if the policy overshoots, it knows how to reorient the object back to the goal orientation. In such a scenario, if the policy overshoots too much, there will be oscillations, and the object will be reoriented back-and-forth around the goal orientation. Empirically, we observe that the policy doesn't show such oscillations as can be seen in 16 new videos on the project website ([https://sites.google.com/view/in-hand-reorientation](https://sites.google.com/view/in-hand-reorientation)) where the goal orientation changes after the previous goal orientation is met (**Continuing Reorientation**). Additionally, we have included 16 new videos where the episode is not terminated after the goal orientation is reached (**No Immediate Termination after Reaching the Goal Orientation**). We can see from these videos that even though the objects are not "still" on reaching the goal orientation, there is no significant overshooting and the objects are largely stable.
>
> With that being said, there is certainly room to improve stability at the goal state. For example, during training, we can set the success criterion to be that the object needs to stay close to the goal orientation for several consecutive time steps or the velocity of the object needs to be 0, etc. These small improvements can be easily incorporated into our work. However, since our setup already works well. We leave it to future work as rerunning all the experiments with such modifications will take more time than the rebuttal period.
>
> **Q: Inconsistent reasoning on “position control is preferred to torque control for sim-to-real transfer”.**
>
> **A:** We are sorry for the confusion. It is common to use PD/PID controllers as part of position control. Due to motor inertia, joint damping affects how well the PID controller works. Since we don't have ground truth access to these parameters in the simulator, we wanted to evaluate the robustness of the learned policies to these parameters. Therefore we perform domain randomization with parameters such as joint damping and tension. Finally, the choice for using position control is guided by our experience in transferring policies on real robots. Many papers [1,2] have shown sim-to-real transfer using position controllers and found that randomizing the dynamics improves sim-to-real transfer performance.
>
> **Q: Point clouds are in general noisy and might contain even more noise than tested in simulation with domain randomization when factoring the occlusions and interaction between the hand and the object.**
>
> **A:** We agree that real-world point clouds can be noisier. As a proxy for real-world noise, we perform noise augmentation on the simulated point cloud and report results as explained in Appendix D.5. We are happy to incorporate specific noise models if the reviewer thinks it would make a stronger case for sim-to-real transfer.
>
> We believe that occlusion will not be a major contributing factor for the difference in the simulated and real point cloud. The reason is that even in simulation, we do not use the ground truth point cloud of the hand or the object, but rather we reconstruct the point cloud using RGBD cameras placed on top of the hand. Therefore, in the simulation, the point cloud also has occlusions due to interaction between the hand and the object.
>
> It is possible that when we do the sim-to-real transfer, we might need to tune more on how much noise we add during training. Due to the ongoing pandemic, we are unable to conduct real robot experiments. Our system is designed using observation and action spaces that we believe can be transferred to the real system, something we will validate in future work.
>
> **Q: It might be worth exploring whether using object keypoints can help reduce some of these effects.**
>
> **A:** We thank the reviewer for the valuable suggestion. Using keypoints to represent the object pose is indeed a good approach to mitigate the symmetry issue. However, it comes with its own set of challenges: it can be less robust to occlusions, and it remains unclear how to generalize keypoints to new objects from different categories. We agree with the reviewer that the use of keypoints is a promising direction for future work.

---

> > ### Author Response · Authors · 2021-08-30
> > **Response to Reviewer UVfU (2)**
> >
> > **Q: "Unable to find the experimental results on the zero-shot training performance on objects weighing up to 500g referenced on line 212"**
> >
> > **A:** We are sorry that the reviewer couldn't find the experiment result. The result is actually in the footnote on Page 5.
> >
> > **Q: Use a table to better explain the object variations in section 3 and 4.1.**
> >
> > **A:** Thanks for the suggestion. We have added a table (Table C3 in the appendix) to explain parameters influencing object variations.
> >
> > **Q: "It would be useful to show the effect of these perturbations with an ablation study showing the effect that each of these transformations (and all of them combined) on a model that is trained without them".**
> >
> > **A:** Performance comparison between the model trained with and without data augmentations is reported in Table D.6 in the appendix. We would have loved to investigate the effect of each data augmentation individually — however, we have not done so due to practical considerations. First, it increases the number of experiments many-folds (number of augmentations x number of random seeds), which is beyond our computational budget. Second, individual augmentations do not slow down training noticeably and the method of combining various augmentations has proven useful in contrastive learning [4,5] and reinforcement learning [6].
> >
> > ---
> >
> > In addition, we have bolded the number of the best-performing method in Table 1 and Table D.5 in our updated manuscript as the reviewer suggested.
> >
> >
> > 1. Peng, X.B., Andrychowicz, M., Zaremba, W. and Abbeel, P., 2018, May. Sim-to-real transfer of robotic control with dynamics randomization. In 2018 IEEE international conference on robotics and automation (ICRA) (pp. 3803-3810). IEEE.
> > 2. Open AI, Andrychowicz, O.M., Baker, B., Chociej, M., Jozefowicz, R., McGrew, B., Pachocki, J., Petron, A., Plappert, M., Powell, G., Ray, A. and Schneider, J., 2020. Learning dexterous in-hand manipulation. The International Journal of Robotics Research, 39(1), pp.3-20.
> > 3. Siekmann, J., Green, K., Warila, J., Fern, A. and Hurst, J., 2021. Blind Bipedal Stair Traversal via Sim-to-Real Reinforcement Learning. arXiv preprint arXiv:2105.08328.
> > 4. Chen, T., Kornblith, S., Norouzi, M. and Hinton, G., 2020, November. A simple framework for contrastive learning of visual representations. In International conference on machine learning (pp. 1597-1607). PMLR.
> > 5. He, K., Fan, H., Wu, Y., Xie, S. and Girshick, R., 2020. Momentum contrast for unsupervised visual representation learning. In Proceedings of the IEEE/CVF Conference on Computer Vision and Pattern Recognition (pp. 9729-9738).
> > 6. Raileanu, R., Goldstein, M., Yarats, D., Kostrikov, I. and Fergus, R., 2020. Automatic data augmentation for generalization in deep reinforcement learning. arXiv preprint arXiv:2006.12862.

---

> > > ### Comment · Reviewer_UVfU · 2021-09-03
> > > **Rebuttal Response**
> > >
> > > Thank you for the thoughtful rebuttal responses to my comments and concerns, and incorporating the suggestions for improved clarity in the shown results. Upon also seeing the responses to questions brought up by the other reviewers, I am updating my recommendation to a strong accept as I feel the main comments from the meta-review have been sufficiently addressed.

---

### Author Response · Authors · 2021-08-30
**Shared response to AC and reviewers**

We thank the AC and the reviewers for their comments and suggestions. We are glad that the reviewers found:

(1) our paper to be tackling challenging/hard tasks (Reviewer Cfuq, Reviewer uUVE).

(2) our analysis/experiments to be thorough and convincing (Reviewer UVfU, Reviewer Cfuq, Reviewer uUVE).

(3) our results to be good, impressive, and strong (Reviewer UVfU, Reviewer uUVE).

(4) our method to be effective, promising, and simple (Reviewer Cfuq, Reviewer uUVE).

(5) the proposed gravity curriculum to be a "considerable feat" (Reviewer UVfU).

We have addressed the AC and the reviewers' concerns individually below. We also want to point out a few updates on the paper as follows：

- We have updated our project website ([https://sites.google.com/view/in-hand-reorientation](https://sites.google.com/view/in-hand-reorientation)) with videos showing how the hand and object move if episodes are not terminated upon success and what if the goal changes upon success.
- We got a small amount of improvement on the vision policy's performance on the mug object and b_toy_airplane by running the experiments longer after the submission. We have highlighted (blue) the new results in Table 3 and Table D.6. These small improvements do not affect conclusions drawn in the paper.
- We have incorporated most writing suggestions in the paper. Some of the major changes are highlighted in blue in the updated paper.

---

### Meta-Review · Area_Chair_wE5T · 2021-08-17

**Recommendation:** Accept (Oral)
**Confidence:** 5

**Metareview:**

This paper presents impressive results on in-hand manipulation using imitation learning techniques in simulation.

The reviewers and area chair found the rebuttal to answer most of the lingering questions and concerns brought up by the first round of reviews.

We found this paper to be an extremely impressive systems paper that I recommend for acceptance.

---

> ### Author Response · Authors · 2021-08-30
> **Response to Area Chair (Related works added)**
>
> We thank the AC for the metareview. A common concern raised by two reviewers was that we terminate the episode when the object is re-oriented to the goal position. We have included videos showing that our method does not suffer from overshooting and the object is largely stable in the goal configuration. We have addressed the remainder of the reviewers' concerns individually and updated the manuscript to incorporate the AC's and reviews' suggestions. In the related work section, we have added the discussion of the papers mentioned by the AC.

---

### Decision · Program_Chairs · 2021-09-13

**Decision:**

Accept (Oral)

**Comment:**

This paper presents impressive results on in-hand manipulation using imitation learning techniques in simulation.

The reviewers and area chair found the rebuttal to answer most of the lingering questions and concerns brought up by the first round of reviews.

We found this paper to be an extremely impressive systems paper that I recommend for acceptance.